# Beyond Accuracy: Evaluating the Reasoning Behavior of Large Language Models - A Survey

**Philipp Mondorf & Barbara Plank**
MaiNLP, Center for Information and Language Processing, LMU Munich, Germany
Munich Center for Machine Learning (MCML), Munich, Germany
{p.mondorf,b.plank}@lmu.de

## Abstract

Large language models (LLMs) have recently shown impressive performance on tasks involving reasoning, leading to a lively debate on whether these models possess reasoning capabilities similar to humans. However, despite these successes, the depth of LLMs' reasoning abilities remains uncertain. This uncertainty partly stems from the predominant focus on task *performance*, measured through shallow accuracy metrics, rather than a thorough investigation of the models' reasoning *behavior*. This paper seeks to address this gap by providing a comprehensive review of studies that go beyond task accuracy, offering deeper insights into the models' reasoning processes. Furthermore, we survey prevalent methodologies to evaluate the reasoning behavior of LLMs, emphasizing current trends and efforts towards more nuanced reasoning analyses. Our review suggests that LLMs tend to rely on surface-level patterns and correlations in their training data, rather than on sophisticated reasoning abilities. Additionally, we identify the need for further research that delineates the key differences between human and LLM-based reasoning. Through this survey, we aim to shed light on the complex reasoning processes within LLMs.

## 1 Introduction

> *"These models are castles in the air. They have no foundations whatsoever."*
> — Jitendra Malik (2021)

Reasoning is an integral aspect of human intelligence and deliberate, rational thought (Holyoak & Morrison, 2005). It allows individuals to draw conclusions from available information and move beyond their current knowledge (Lohman & Lakin, 2011). As such, reasoning plays a fundamental role in problem-solving and decision-making, and has been a long-standing goal within the field of artificial intelligence (Robinson & Voronkov, 2001).

In recent years, large language models have demonstrated remarkable performance on tasks that require reasoning (Bubeck et al., 2023; Wei et al., 2022; Kojima et al., 2022). This has sparked a vigorous debate about the extent to which these models possess reasoning abilities similar to humans (Mitchell & Krakauer, 2023; Mitchell, 2023; Borji, 2023). While proponents argue that reasoning capabilities *emerge* with scale, referring to LLMs as *foundation* models (Bommasani et al., 2021; Kaplan et al., 2020), skeptics contend that the models' performance primarily reflects their capacity to memorize the vast amount of data they have been trained on (Wu et al., 2024; Dziri et al., 2023; Razeghi et al., 2022; Zhang et al., 2023). Thus, the question arises: are these models simply *"castles in the air"* with *"no foundations whatsoever,"* as Malik (2021) once stated, or do they possess genuine reasoning capacities? One of the major challenges in this debate is the immense size, complexity, and closed-source nature of popular LLMs and their underlying training data. Moreover, the focus often lies on the models' *performance* on downstream reasoning tasks (Fu et al., 2023; Liu et al., 2023), overshadowing in-depth analyses of their reasoning *behavior*. In this study, we seek to shed light on the ongoing debate by providing a comprehensive overview of research that goes

beyond task accuracy, offering more nuanced insights into the models' reasoning processes. Specifically, we address the following research questions:

RQ1:  How do current LLMs behave across diverse reasoning tasks?

RQ2:  What are the prevalent evaluation methods employed to assess the reasoning behavior of large language models?

**Other Surveys.**   While various surveys on reasoning within large language models have emerged in recent years (Huang & Chang, 2023; Yu et al., 2024; Sun et al., 2024; Yang et al., 2023; Qiao et al., 2023; Chu et al., 2023; Mialon et al., 2023; Liu et al., 2024; Ahn et al., 2024), these studies predominantly focus on techniques that seek to enhance or benchmark the *performance* of LLMs on downstream reasoning tasks. However, there seems to exist no review of work yet that assesses the reasoning *behavior* of LLMs, rather than their final task performance. Hence, the contributions of this survey are as follows: (i) To the best of our knowledge, we present the first comprehensive review of literature that evaluates LLM-based reasoning beyond task accuracy, offering deeper insights into the models' reasoning *behavior*, and (ii) we suggest a taxonomy that categorizes prevalent evaluation methods designed to analyze the reasoning dynamics of LLMs.

## 2   Terminology

We begin this survey with an introduction to fundamental concepts and terminologies pertinent to reasoning in both humans and LLMs. The study of reasoning has captivated scholarly interest for centuries, with insights from diverse disciplines such as philosophy, logic, psychology, neuroscience and artificial intelligence (Holyoak & Morrison, 2005). Over the years, various definitions of reasoning have been proposed, with each discipline offering its unique perspective. In this survey, we define reasoning in the broadest sense (Leighton, 2003; Holyoak & Morrison, 2005; Byrne et al., 1993), and direct the interested reader to the work by Yu et al. (2024) and Sun et al. (2024) for more domain-specific definitions.

**Definition 2.1** (Reasoning). *The process of drawing conclusions based on available information (usually a set of premises).*

It is imperative to note that reasoning is a fundamentally process-oriented activity, rather than a singular endpoint (Leighton, 2003; Johnson-Laird, 2006). Although this process often remains hidden in humans, manifesting predominantly through the final conclusions inferred, cognitive psychology leverages methodologies like "think aloud" protocols to unveil the cognitive mechanisms underpinning reasoning (Wolcott & Lobczowski, 2021; Van der Henst et al., 2002; Rips, 1989). Similarly, to understand the reasoning capabilities of large language models, it is crucial to consider not merely the end result, but the reasoning process itself. Thus, we differentiate between reasoning *behavior* and reasoning *performance*. Drawing from behavioral psychology, which views behavior as an organism's response to stimuli (American Psychological Association, n.d.), we define reasoning behavior as follows:

**Definition 2.2** (Reasoning Behavior). *The system's computed response to a reasoning task (the stimulus), particularly its actions, expressions and underlying mechanisms exhibited during the reasoning process.*

Our working definition highlights the procedural aspects of reasoning, rather than its final outcome. Understanding a model's reasoning behavior involves analyzing *how* it arrives at its conclusions. Conversely, reasoning *performance* is outcome-oriented. It is typically measured in terms of accuracy or the efficiency with which relevant conclusions are drawn (Barredo Arrieta et al., 2020). While performance can be helpful in evaluating a system's capacities to tackle specific reasoning tasks, an analysis of reasoning *behavior* can yield deeper insights into the process itself, thereby offering a more comprehensive understanding.

### 2.1   A Categorization of Reasoning Tasks

Analogous to the assessment of human reasoning within the field of cognitive psychology (WOOD, 1969; Byrne et al., 1993; Dewey, 2022), the evaluation of reasoning capabilities in

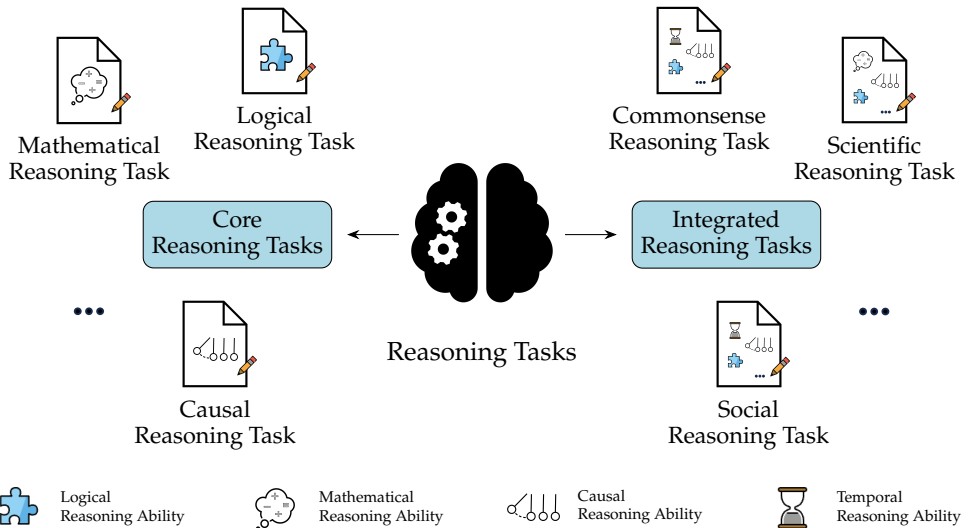

Figure 1: Schematic overview of the two types of reasoning tasks distinguished in this survey. *Core* reasoning tasks are designed to assess a particular reasoning ability within a given context. Conversely, *integrated* reasoning tasks involve the concurrent use of various reasoning skills. Tasks and abilities listed are not exhaustive.

large language models predominantly occurs through their engagement with designated reasoning tasks (Sun et al., 2024). These tasks are designed to elicit the system's capability of drawing conclusions relevant to the problem at hand. In this survey, we distinguish between *core* and *integrated* reasoning tasks. *Core* reasoning tasks are designed to assess fundamental reasoning skills in an isolated manner. They typically aim to test a single type of reasoning, such as logical, mathematical or causal reasoning. Examples of such tasks may include syllogisms, basic arithmetic problems or structured causal-graph predictions. Conversely, *integrated* reasoning tasks require the concurrent use of various reasoning types, thereby assessing a combination of fundamental reasoning skills. Examples are commonsense or scientific reasoning tasks. Such problems often reflect the complex cognitive challenges humans encounter in everyday life and professional settings. A schematic representation distinguishing these task categories is provided in Figure 1.

For the purpose of this survey, our focus is confined to examining the behaviors of LLMs in the context of *core* reasoning tasks, specifically logical, mathematical, and causal reasoning tasks. We leave a review of the models' reasoning behaviors within the context of *integrated* reasoning tasks to future work.

## 3 Reasoning Behavior of LLMs

This section reviews studies that extend beyond mere task accuracy, focusing instead on evaluating the reasoning *behavior* of large language models.[1] Through this review, we intend to address RQ1 and shed light on how these models currently behave across three *core* reasoning tasks: logical reasoning (Section 3.1), mathematical reasoning (Section 3.2), and causal reasoning (Section 3.3).

---

[1] We consistently specify when a particular technique, other than standard prompting, is utilized. For a review of prompting approaches designed to improve the models' reasoning performance, we recommend consulting prior surveys by Huang & Chang (2023), Yu et al. (2024), or Sun et al. (2024).

### 3.1 Behavior in Logical Reasoning Tasks

The study of logical reasoning evolves around the question of how individuals infer valid conclusions from a set of given premises within a structured framework of logical rules and principles (Holyoak & Morrison, 2005). Based on how conclusions are inferred, logical reasoning can be classified into three types: *deductive*, *inductive*, and *abductive* reasoning (Johnson-Laird, 2008; Flach & Kakas, 2000). While in *deductive* reasoning, conclusions necessarily follow from the premises' truth, *inductive* and *abductive* reasoning are considered defeasible, i.e. conclusions are at most probable, but never necessary (Koons, 2022). *Inductive* reasoning is concerned with deriving general conclusions from specific instances, whereas *abductive* reasoning entails formulating plausible hypotheses that explain the data observed. For additional details on the distinction between the three types of logical reasoning, we refer to prior surveys on reasoning performance within LLMs, such as the ones by Yu et al. (2024) or Sun et al. (2024).

#### 3.1.1 Behavior in Deductive Reasoning Tasks

Extensive research has been dedicated to examining the reasoning behavior of LLMs in the context of deductive reasoning tasks. Various studies analyze the validity and consistency of logical proofs generated by LLMs within deductive reasoning problems. For instance, Saparov & He (2023) conduct a systematic assessment of the rationales produced by various GPT-3 iterations (Brown et al., 2020), through chain-of-thought (CoT) prompting (Wei et al., 2022). By parsing each step of the generated reasoning trace into first-order-logic, its *validity*, *atomicity*, and *utility*[2] is measured. Findings indicate that, in comparison to their smaller counterparts, larger models are more adept at generating both valid and atomic steps. However, the utility of these steps is often low, resulting in misleading steps from which the models struggle to recover. In a similar vein, Dziri et al. (2023) study rationales of LLMs by parsing them into computation graphs, where each node represents a partial solution of the multi-step reasoning process. Evaluations on Einstein's puzzle (Vassberg & Vassberg, 2009) indicate that GPT-3, ChatGPT (OpenAI, 2022) and GPT-4 (OpenAI et al., 2024) solve the task by reducing multi-step reasoning into shortcut pattern matching. This shortcut behavior can yield correct answers when the model has been exposed to similar patterns during training, but falls short in generalizing to novel or more complex instances. Furthermore, theoretical findings suggest that due to the models' autoregressive nature, mistakes in early stages of the reasoning process can lead to compounding errors that exponentially diminish the likelihood of arriving at the correct solution in subsequent steps.

Further studies highlight that LLMs tend to rely on superficial statistical features, rather than engaging in systematic reasoning. Chen et al. (2024b) illustrate that the premise order markedly influences the LLMs' behavior in propositional reasoning tasks. Specifically, when premises are presented in an order that does not align with the ground-truth proof, models such as ChatGPT, GPT-4, PaLM 2-L (Anil et al., 2023) and Gemini Pro (Team et al., 2023) encounter significant difficulties within their reasoning, even though such an ordering does not change the underlying logic. Zhang et al. (2023) show that an over-reliance on statistical features in the training data can hinder a model's reasoning and generalization capacity. By eliminating certain statistical cues from its training dataset, BERT (Devlin et al., 2019) demonstrates enhanced generalization capabilities in propositional logic. Similarly, Pirozelli et al. (2023) indicate that various fine-tuned language models show difficulties in transferring their logical reasoning ability when cross-probed on unseen deductive reasoning tasks.

Additional research points to the difficulties of LLMs in understanding specific logical operators. Sanyal et al. (2022) show that GPT-3, RoBERTa (Liu et al., 2019), and models from the T5 series (Raffel et al., 2020) exhibit deficiencies in comprehending logical negations, often failing to correctly deduce the implications of negated statements and rules. Similar findings have been reported by Truong et al. (2023), who demonstrate that models lack a sensitivity to negations within a broad range of natural language inference tasks. Wan

---

[2]*Validity* denotes whether the proof step logically follows from preceding steps, *atomicity* reflects whether it can be proven with exactly one application of a deduction rule, and *utility* measures its direct contribution towards the derivation of the final conclusion.

et al. (2024) comprehensively assess a suite of formal reasoning scenarios, subjecting GPT-3, ChatGPT, GPT-4, Bard (PaLM 2), Vicuna (Vicuna, 2023) and Guanaco (Dettmers et al., 2023) to minimum functionality tests (MFTs), serving as logical reasoning unit tests that gauge the models' inherent logic comprehension. Their analysis uncovers a common difficulty among models in identifying logical fallacies. In addition, GPT-4 appears to struggle with De Morgan's Laws, which relate conjunctions and disjunctions through logical negation.

A growing body of research investigates the extent to which LLMs exhibit human-like reasoning patterns, particularly in deductive reasoning tasks. For example, Eisape et al. (2024) explore the parallels between human reasoning and that of LLMs in syllogisms. Their findings suggest that LLMs, much like their human counterparts, are susceptible to common logical fallacies and cognitive biases. Similarly, Dasgupta et al. (2022) find that LLMs, akin to humans, display content effects, indicating that the problem's semantic content can significantly influence the models' reasoning behavior. Further research corroborates the manifestation of human reasoning patterns in LLMs, as outlined in Appendix A.

**Mechanistic Evaluation.** Additional research evaluates the reasoning behavior of LLMs by inspecting the models' internal mechanisms during the reasoning process. For example, Hou et al. (2023) analyze the models' attention patterns, specifically those of GPT-2 (Radford et al., 2019) and LLaMA (Touvron et al., 2023), to uncover if and how these models perform multi-step reasoning internally. Findings indicate a structured, step-wise information processing within the models. Furthermore, layer-wise probing reveals that LLMs tend to focus on identifying relevant information from the task in lower layers, transitioning to more intricate reasoning processes in higher layers. Pirozelli et al. (2023) similarly probe individual layers of a fine-tuned RoBERTa-large model, corroborating the pivotal role of higher layers in the reasoning process. Dutta et al. (2024) present an in-depth investigation into the internal mechanisms of LLaMA 2-7B (Touvron et al., 2023) when instructed via chain-of-thought prompting. Findings indicate a notable functional rift in the model's middle layers, where token representations in the initial half are biased towards pre-training priors, and an abrupt shift to in-context priors occurs in the latter half. As opposed to the findings by Hou et al. (2023), the study suggests that LLaMA 2-7B employs multiple, concurrent pathways, instead of following a singular path of reasoning.

### 3.1.2 Behavior in Inductive Reasoning Tasks

In contrast to the well-examined domain of deductive reasoning, the reasoning behavior of LLMs in inductive reasoning tasks remains comparatively underexplored. Nevertheless, research exists that seeks to evaluate the capability of LLMs to infer general conclusions from specific examples. For instance, Yang et al. (2024) investigate how LLMs, such as GPT-J (Wang & Komatsuzaki, 2021) and LLaMA 7B, induce general rules from given facts. While in principle the models seem able to infer general rules from the data provided, challenges arise in ensuring that the rules are consistent with the premises, extend beyond the given information, align with real-world knowledge, and are pertinent to the given task. Similarly, Qiu et al. (2024) find that LLMs such as GPT-3.5, GPT-4, Claude 2 (Anthropic, 2023), and LLaMA 2-70B are capable of inferring rules from given data. However, the models frequently err in the application of these rules, highlighting a gap between their ability to generate and apply rules. Moreover, the rules derived often diverge significantly from those humans might produce, exhibiting a tendency towards verbosity and an inability to concentrate on the fundamental patterns for generalization. In addition, the study finds that LLMs display a pronounced sensitivity to alterations of the task descriptions. Han et al. (2024) examine the reasoning behaviors of GPT-3.5 and GPT-4 in property induction tasks, where properties common among different categories need to be induced. Findings suggest that GPT-4's behavior closely aligns with human judgments. However, challenges arise when the models need to handle premise non-monotonicity, a scenario in which adding more information to an argument can actually decrease its likelihood. For instance, Han et al. (2024) illustrate that the three-premise argument {crow, peacock, rabbit} → bird is considered weaker by humans than the two-premise argument {crow, peacock} → bird, a line of reasoning that both GPT-3.5 and GPT-4 seem to struggle with.

### 3.1.3 Behavior in Abductive Reasoning Tasks

Some efforts have been made to evaluate the behavior of LLMs in abductive reasoning tasks. Collins et al. (2022) ask both humans and GPT-3 to generate plausible explanations for unexpected counterfactual scenarios, for instance why a window did not break despite being struck by a rock. Analyses indicate that GPT-3's ability to generate plausible and coherent explanations for scenarios that require reasoning beyond established patterns is limited. In scenarios demanding inventive, coherent, and context-sensitive responses, especially when the usage of common nouns is restricted, human performance distinctly surpasses that of GPT-3, underscoring a pronounced deficiency in the model's ability to reason beyond its training distribution. Further, Rudinger et al. (2020) reveal that language models such as GPT-2, BART, and various T5 variants exhibit difficulties in identifying or generating arguments that either weaken or strengthen a given hypothesis. Even after fine-tuning, models tend to struggle with the task, often contradicting themselves by producing identical statements to strengthen and weaken the same premise-hypothesis pair. Xu et al. (2023) explore the behavior of GPT-3.5, ChatGPT, and PaLM 2 on various abductive reasoning tasks, paying particular attention to the models' rationales and errors manifested during reasoning. The paper highlights a tendency for LLMs to incorporate redundant information in their explanations and to generate content not grounded in the context, resulting in hallucinations. In comparison to deductive reasoning scenarios, findings suggest that models seem to particularly struggle with multi-hop reasoning in abductive reasoning tasks.

### 3.2 Behavior in Mathematical Reasoning Tasks

Mathematical reasoning encompasses the structured process of arriving at conclusions based on established mathematical principles and logical deduction (Horsten, 2023). Several studies explore the behavior of LLMs in the context mathematical reasoning tasks, especially in arithmetic and math word problems (MWPs). For instance, Srivastava et al. (2024) evaluate a set of LLMs on a *functional* variant of the MATH dataset (Hendrycks et al., 2021), where the underlying mathematical principles of each problem are captured rather than a static problem formulation. This allows for evaluating models on different dataset *snapshots*, each comprising unique problem formulations but the same underlying reasoning process. Results reveal inconsistencies across varying snapshots, indicating a tendency of models to rely on memorization rather than reasoning. Razeghi et al. (2022) further highlight the impact of how a problem is formulated, noting a marked correlation between the frequency of terms in the models' pre-training data and their ability to solve arithmetic tasks. Similarly, Wu et al. (2024) observe that models struggle with two-digit additions expressed in mathematical bases that are less represented in the models' training data.

Further studies underline the models' susceptibility to perturbations of the reasoning task. Shi et al. (2023) find that LLMs such as `code-davinci-002` and GPT-3.5 can be distracted by context irrelevant to the MWP solution, especially when the irrelevant context shares lexical similarities with the original problem formulation. In a similar vein, Stolfo et al. (2023) report that LLMs are sensitive to interventions on MWPs, such as changing numerical values or altering the textual framing of the problem.

Other studies investigate whether human-like reasoning behavior in mathematical tasks manifests in LLMs. Hagendorff et al. (2023) show that, akin to humans, GPT-3 variants tend to offer intuitive yet incorrect answers to cognitive reflection tests (Frederick, 2005). Notably, GPT-3.5 and GPT-4 provide more deliberate responses, outperforming humans in avoiding intuitive errors. At the same time, McKenzie et al. (2023) demonstrate a form of goal misgeneralization, where models, tasked with rounding numbers to a specific number of significant digits, consistently round based on the number of decimal places. This reflects a cognitive bias known as attribute substitution (Morewedge & Kahneman, 2010), where a more challenging task is replaced with a simpler, related task. In a mechanistic evaluation, Chen et al. (2024a) conduct a layer-wise analysis of LLaMA's mathematical reasoning capabilities, discovering that higher layers exhibit superior mathematical problem-solving abilities, while lower layers seem to lack basic arithmetic and factual knowledge.

### 3.3 Behavior in Causal Reasoning Tasks

Causal reasoning is the process of discerning the cause-and-effect relationships that govern the dynamics of our environment (Sloman, 2005). Extending beyond correlation, it offers a more nuanced understanding of how changes in one variable can bring about changes in another (Pearl, 2009). For a comprehensive introduction to causal reasoning, we strongly recommend the work of Pearl (2009). While research on the behavior of LLMs in causal reasoning tasks is still in its early stages, the field is receiving growing attention. Various studies to date suggest that LLMs are capable of reciting causal facts from their training data, but lack an intrinsic ability to comprehend or construct causal relationships. For instance, Zečević et al. (2023) indicate that while LLMs, including GPT-3, Meta AI's Opt, and AlephAlpha's Luminous (AlephAlpha, 2022), demonstrate some proficiency in causal reasoning tasks that align with causal facts seen during training, they exhibit limited ability to accurately discern and apply causal relationships in scenarios that demand more than associative recalls from their training data. Jin et al. (2023) probe the models' behavior across three levels of causation: (1) the *associational*, (2) the *interventional*, and (3) the *counterfactual*, as outlined by Pearl & Mackenzie (2018)'s *Ladder of Causation*. Similarly, their findings indicate that LLMs are more adept at answering *associational* queries than tackling *interventional* or *counterfactual* tasks. Models seem to particularly struggle with causal relationships that deviate from commonsense or are unlikely to be part of their training corpora. Jin et al. (2024) evaluate the ability of LLMs to infer causation from statements that describe correlations between variables. Analyses reveal significant challenges in solving the given task across seventeen LLMs. Although fine-tuning offers substantial improvements, models still fail to generalize in out-of-distribution scenarios. Kosoy et al. (2023)'s evaluation of GPT-3 and PaLM on the *blicket detector* task (Gopnik & Sobel, 2000), where models need to infer which objects cause a light to switch on, further highlights challenges in causal reasoning with LLMs. While models can identify the correct causal structure when a set of causal hypotheses is provided, they struggle in conditions where the hypotheses are not given, underscoring a limitation in their ability to infer causal relationships from limited data.

**Behavior in Counterfactual Reasoning Tasks.** Positioned at the last level of Pearl & Mackenzie (2018)'s *Ladder of Causation*, counterfactual tasks assess the models' ability to reason about hypothetical scenarios. Research into the behavior of LLMs in counterfactual scenarios reveals notable challenges in current models. Studies, like those conducted by Frohberg & Binder (2022), focus on the ability of LLMs to predict outcomes in hypothetical setups, identifying a significant gap between the capabilities of humans and LLMs, particularly in highly unrealistic scenarios. Li et al. (2023) further illustrate that while models like GPT-3 can produce outcomes that seem to align with counterfactual propositions, these models heavily rely on simple lexical cues within the given context, rather than demonstrating a profound grasp of the scenarios' hypothetical essence. Yu et al. (2023) evaluate LLMs on questions embedded in counterfactual presuppositions, pushing the models beyond simple fact retrieval. Their findings suggest that "closed-book" models such as GPT-3 and `code-davinci-002` may fabricate facts or base their responses on flawed premises when answering counterfactual questions. Huang et al. (2024) explore a different angle by asking models to modify a given piece of argumentative text so that it upholds a specified logical relationship under new premises. Findings indicate that while LLMs like GPT-3.5 and GPT-4 demonstrate a capacity for solving such tasks, they struggle with modifying arguments such that a new premise makes the argument less likely to be true, an observation also made by Rudinger et al. (2020) in the context of abductive reasoning.

**Summary.** Regarding RQ1, our review suggests that the reasoning behaviors of LLMs are nuanced and diverse, yet a significant trend emerges: while current LLMs demonstrate proficiency in reasoning problems that align with their training distribution, they frequently encounter substantial conceptual difficulties in out-of-distribution scenarios. Notably, slight alterations in the task context can markedly impair the models' reasoning capabilities. Multi-step reasoning is often reduced to shortcut pattern matching, and fundamental conceptual errors highlight the models' deficiency in understanding basic principles of logic, mathematics, and causal reasoning, particularly in counterfactual setups.

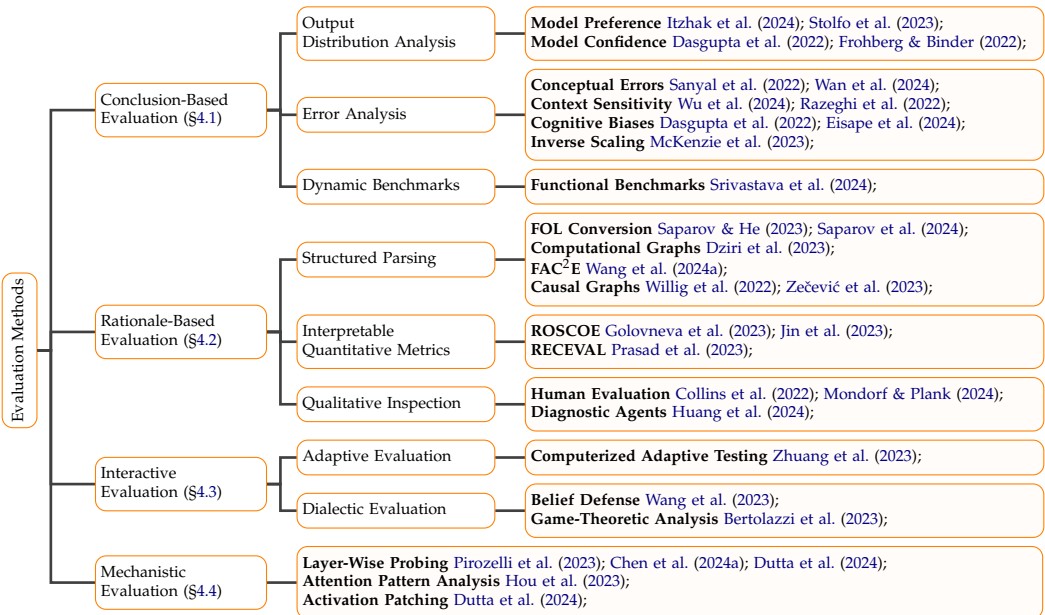

Figure 2: A taxonomy of evaluation methods to analyze the reasoning behaviors of LLMs. Only representative approaches for each method are listed.

## 4 Evaluation Methods

So far, we have provided an overview of studies that evaluate the behavior of LLMs on three core reasoning tasks. However, to date, a standardized methodology for assessing the reasoning capabilities of large language models is lacking. To address RQ2, we review and categorize predominant evaluation frameworks for analyzing the *behavior* of LLMs in reasoning tasks. As depicted in Figure 2, we categorize evaluation methodologies into four distinct groups: (i) *conclusion-based*, (ii) *rationale-based*, (iii) *interactive*, (iv) and *mechanistic* evaluations. In the following sections, we further delineate each category by discussing prevalent techniques. An overview of the advantages and disadvantages of each evaluation procedure can be found in Table 1. Additional details are provided in Appendix B.

### 4.1 Conclusion-Based Evaluation

In conclusion-based evaluation schemes, emphasis is placed on the model's final answer rather than the process by which the model arrives at its conclusion. This outcome-oriented approach, despite its limitation of overlooking the model's rationales, can nonetheless provide insights into the model's reasoning behavior. For instance, a thorough *error analysis* can unveil conceptual errors (Sanyal et al., 2022), reflections of cognitive biases (Dasgupta et al., 2022), or sensitivities to the task context (Wu et al., 2024). Similarly, an examination of the model's *output distribution* may reveal inherent predispositions towards certain outcomes (Itzhak et al., 2024), or serve as an indicator of the model's confidence in specific conclusions (Frohberg & Binder, 2022). However, relying solely on the model's final conclusion can be less reliable than also considering how the model arrives at its conclusion. For example, candidate answers derived from first token probabilities in multiple-choice setups are often not robust (Wang et al., 2024b), and final answers might not always align with the model's verbalized reasoning (Mondorf & Plank, 2024; Ye & Durrett, 2022). Moreover, answers to benchmark datasets might have been compromised by data leakage during the model's training process, limiting the insights that can be drawn from a correct conclusion (Balloccu et al., 2024; Xu et al., 2024). To address the issue of data contamination and test the model's capacity to generalize, *dynamic benchmarks* can be employed that update over time. For instance, functional benchmarks capture the underlying reasoning process rather than a

static problem formulation, probing the model's reasoning capabilities on dataset snapshots released quarterly (Srivastava et al., 2024).

## 4.2 Rationale-Based Evaluation

In contrast to conclusion-based evaluation schemes, rationale-based evaluation methods focus on examining the reasoning trace generated by the model, typically assessing its logical validity and coherence. While this approach allows for more nuanced insights into the model's reasoning behavior, rationale-based evaluation schemes are often more challenging to automate and scale. Given the variability in the model's reasoning, a spectrum of assessment techniques exist. For rationales following a highly consistent format—either through structured contexts or methods that guide the model's response style (Wan et al., 2024)—rationales can be *parsed* into more formalized representations such as first-order logic (Saparov & He, 2023), computation graphs (Dziri et al., 2023), or causal graphs (Willig et al., 2022), facilitating a more granular examination. Alternatively, *interpretable quantitative metrics*, such as ROSCOE (Golovneva et al., 2023) or RECEVAL (Prasad et al., 2023), may be utilized to evaluate the rationales' semantic alignment with the reasoning task, their coherence, factual consistency, and logical validity. In instances where rationales are less structured, qualitative inspections are commonly employed, either through diagnostic agents (Huang et al., 2024), or human judgments (Mondorf & Plank, 2024).

## 4.3 Interactive Evaluation

Similar to the principle of dynamic assessment within psychology (Haywood & Tzuriel, 2002), interactive evaluation offers a framework to engage with LLMs during the evaluation. This approach allows for more flexible assessments tailored to the model's specific reasoning behavior. Although such evaluations are often costly and challenging to scale, several techniques have been developed to automate the process. For instance, *adaptive evaluations* dynamically select reasoning tasks based on the model's responses, thus providing deeper insights into its capabilities and limitations beyond what static questionnaires can reveal (Zhuang et al., 2023). *Dialectic evaluation* methods assess the model's reasoning in dialogue form, for example, by challenging the model's conclusions (Wang et al., 2023), or engaging it in game-theoretical scenarios (Bertolazzi et al., 2023). While interactive evaluations yield nuanced insights, they lack the structure of traditional methods, posing challenges in terms of standardization and reproducibility.

## 4.4 Mechanistic Evaluation

Mechanistic evaluations of LLMs delve into the underlying processes that drive the model's responses, aiming to uncover the *"how"* and *"why"* within their reasoning processes. By analyzing the internal mechanisms such as attention patterns (Hou et al., 2023), activation flows (Dutta et al., 2024), and the functional attributes of individual layers (Pirozelli et al., 2023), deeper insights into the model's operational logic can be gained, as illustrated in Section 3.1.1. Focusing on the model's intrinsic processes, this framework contrasts with previous approaches, drawing parallels to the study of human reasoning from a neuroscientific perspective (Papo, 2015). Nonetheless, current methods remain compute-intensive, and their findings may not always generalize across different models and tasks (Bereska & Gavves, 2024; Ferrando et al., 2024).

# 5 Discussion

Despite the notable performance of large language models in prominent reasoning tasks (Bubeck et al., 2023; Fu et al., 2023), our review suggests that current models more closely resemble *stochastic parrots* (Bender et al., 2021) than systematic reasoners. As discussed in Section 3, we find that although many LLMs demonstrate proficiency in reasoning problems that align with their training data, the models' reasoning behavior reveals significant conceptual errors and limitations in out-of-distribution scenarios. As highlighted by Mahowald et al. (2024), this suggests a limited functional linguistic competence in LLMs. It is likely

| Evaluation Method | Advantages | Disadvantages |
|---|---|---|
| **Conclusion-based evaluation** | Allows for controlled setups
Provides metrics for comparison
Easy to automate and scale
Easy to reproduce | Limited insights
Less reliable |
| **Rationale-based evaluation** | Offers more nuanced insights
More robust in certain scenarios | Difficult to automate and scale
Might require expert interpretation |
| **Interactive evaluation** | Highly flexible
Customizable to model behavior | Expensive
Difficult to automate and scale
Less standardized and reproducible |
| **Mechanistic evaluation** | Identifies features or circuits responsible for specific behaviors
Supports direct interventions on model internals | Findings may not generalize across tasks or models
Results may be hard to interpret
Compute-intensive |

Table 1: Comparison of strengths and limitations of the different evaluation methods.

that the apparent success of LLMs in reasoning tasks predominantly reflects their ability to memorize the extensive data they have been trained on (Wu et al., 2024; Dziri et al., 2023). Recent studies indicate that a substantial amount of benchmark datasets has been leaked to current LLMs (Balloccu et al., 2024; Xu et al., 2024), raising concerns about the insights derived from their performance on such benchmarks. Therefore, we advocate for more nuanced analyses of the models' reasoning behavior, particularly in novel scenarios that the models have not previously encountered.

While human reasoning is not infallible (Johnson-Laird, 2010), the human capacity for robust reasoning and generalization from limited data remains unmatched by current LLMs. Various studies point to the fundamental differences between human reasoning and that of LLMs, especially the models' restrictive autoregressive pre-training objective (McCoy et al., 2023; Shanahan, 2024; Lenci, 2023). We call for further research—particularly on reasoning behavior—within both humans and LLMs to better discern and comprehend the essential components missing in LLMs, which are crucial for robust and systematic reasoning.

## 6 Conclusion

This survey provides a comprehensive review of literature that evaluates LLM-based reasoning beyond mere task accuracy, offering deeper insights into the reasoning behavior of large language models. We discuss the behavior of LLMs across three core reasoning tasks (RQ1), assessing logical, mathematical and causal reasoning. Furthermore, we outline predominant evaluation frameworks and compare their respective strengths and limitations (RQ2). Our findings indicate that although LLMs demonstrate proficiency in reasoning problems that align with their training data, they often encounter significant conceptual challenges in out-of-distribution scenarios. Considering these insights and the recent issue of benchmark dataset leakage (Balloccu et al., 2024; Xu et al., 2024), we caution against relying solely on shallow accuracy metrics for static reasoning tasks. Instead, we advocate for more sophisticated reasoning assessments, such as those discussed in Section 4.

**Acknowledgments**

We express our gratitude to the anonymous reviewers for their insightful comments and suggestions. Furthermore, we would like to acknowledge that the emojis featured in Figure 1 are designed by OpenMoji – the open-source emoji and icon project (License: CC BY-SA 4.0). Finally, we gratefully recognize the support for BP through the ERC Consolidator Grant DIALECT 101043235.

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

# A  Human-Like Reasoning Behavior in LLMs

Recent research has begun to explore the extent to which human reasoning behaviors are reflected in large language models, given their training on human-generated data. Various studies delve into the manifestation of human-like reasoning behaviors within LLMs in the context of deductive reasoning tasks. For instance, Eisape et al. (2024) compare the behaviors of humans and LLMs, particularly those from the PaLM 2 series, in syllogistic reasoning tasks. The findings reveal that, similar to humans, LLMs are prone to logical fallacies and cognitive biases such as ordering effects. This susceptibility persists across model sizes, though larger models tend to engage in more deliberate reasoning, showing a reduced sensitivity towards these errors. A notable finding indicates that LLMs, unlike humans, rarely produce the "nothing follows" response, even when it represents the accurate deduction. In a similar vein, Dasgupta et al. (2022) demonstrate that LLMs, akin to humans, exhibit content effects in predicate logic problems, i.e. their reasoning is influenced by the semantic content of the task. This effect is evident in various LLMs including ChatGPT, PaLM 2, and Chinchilla (Hoffmann et al., 2022). In particular, analyses indicate that the models' reasoning in syllogisms is biased by the believability of the conclusion, a behavior known as belief bias (Klauer et al., 2000). Further research corroborates the influence of semantic content on the models' reasoning. Ando et al. (2023) demonstrate a belief-bias in models like ChatGPT, RoBERTa and BART (Lewis et al., 2020) by comparing the models' behavior on abstract, belief-consistent, and belief-inconsistent syllogisms. Their analyses also uncover a predisposition towards conversion errors and atmosphere effects, where logical quantifiers are either misinterpreted or lead to uninformed inferences, respectively (Tversky & Kahneman, 1974). Itzhak et al. (2024) further show that methods such as instruction tuning and reinforcement learning from human feedback (RLHF) (Ouyang et al., 2022) may amplify cognitive biases within LLMs. Mondorf & Plank (2024) delve into the inferential strategies of open-access LLMs in problems of propositional logic. Comprehensive manual evaluations of the models' rationales reveal that LLMs utilize inferential strategies similar to those employed by human reasoners. Their evaluations further underline difficulties of LLMs with logical negations and a vulnerability to logical fallacies commonly observed in human reasoning. Similarly, McKenzie et al. (2023) indicate that LLMs tend to replicate human-like logical errors when engaging with the logical principle of modus tollens, suggesting an imitation of flawed reasoning patterns from their training data. Notably, this trend becomes more pronounced with increasing model size, a phenomenon denoted as *inverse scaling*.

# B  Further Details on Evaluation Methods

In this section, we offer a more detailed overview of the evaluation frameworks commonly employed to examine the reasoning behavior of large language models. Expanding on the taxonomy depicted in Figure 2, we present *conclusion-based* evaluation methods in Table 2, provide an overview of *rationale-based* evaluation approaches in Table 3, discuss *interactive* evaluation techniques in Table 4, and delineate *mechanistic* evaluation strategies in Table 5.

| Category | Evaluation Method | Description | Exemplary Reference |
|---|---|---|---|
| Output Distribution Analysis | Model Preference | Evaluate the likelihood of various candidate answers from a predefined set to assess the model's tendency towards specific answers. | Itzhak et al. (2024) |
| | Model Confidence | Interpret the probability assigned to an answer as a measure of confidence towards that answer. | Dasgupta et al. (2022) |
| Error Analysis | Conceptual Errors | Assess the model's errors with respect to fundamental principles and concepts within the domain of reasoning. | Sanyal et al. (2022) |
| | Context Sensitivity | Evaluate the model's robustness towards perturbations of the task's context. | Wu et al. (2024) |
| | Cognitive Biases | Probe the model with respect to cognitive biases or heuristics commonly encountered in human reasoning. | Eisape et al. (2024) |
| | Inverse Scaling | Analyze scenarios in which larger models tend to exhibit more pronounced errors than smaller models. | McKenzie et al. (2023) |
| Dynamic Benchmarks | Functional Benchmarks | Transform static benchmarks in question-answer format into functional form, where triplets of (problem, solution, input) are defined that represent the underlying reasoning process of the task, and thus allow to produce flexible question-answer pairs. | Srivastava et al. (2024) |

Table 2: Overview of conclusion-based evaluation methods that assess the model's reasoning behavior by focusing on the final answers they produced.

| Category | Evaluation Method | Description | Exemplary Reference |
|---|---|---|---|
| Structured Parsing | FOL Conversion | Translate the model's rationale into first-order-logic (FOL) language and evaluate the logical expressions. | Saparov & He (2023) |
| | Computational Graphs | Parse the rationale into a computation graph where vertices represent intermediate results and edges denote function mappings. | Dziri et al. (2023) |
| | FAC$^2$E | Use capability-specific instructions to elicit intermediate structured reasoning steps (crystallized and fluid reasoning). Evaluate each step separately, using ground truth responses and the BARTScore-Recall metric. | Wang et al. (2024a) |
| Interpretable Quantitative Metrics | ROSCOE | Assess the model's rationale using a suite of nuanced metrics, quantifying its semantic alignment, semantic similarity, logical inference, and language coherence. | Golovneva et al. (2023) |
| | ReCEval | Analyze the model's rationale using metrics that quantify its intra-step and inter-step logical validity, as well as the informativeness of each reasoning step. | Prasad et al. (2023) |
| Qualitative Inspection | Human Evaluator | Inspect rationales manually through human annotators. | Mondorf & Plank (2024) |
| | Diagnostic Agents | Assess rationales by means of additional language models, acting as diagnostic agents. | Huang et al. (2024) |

Table 3: Rationale-based evaluation methods that examine the model's reasoning behavior through an analysis of its underlying rationale.

| Category | Evaluation Method | Description | Exemplary Reference |
|---|---|---|---|
| Adaptive Evaluation | Computerized Adaptive Testing (CAT) | Dynamically adapt questions presented during the evaluation procedure based on the model's responses. | Zhuang et al. (2023) |
| Dialectic Evaluation | Belief Defense | Test the model's reasoning through challenging its prior response in a conversational discourse. | Wang et al. (2023) |
| | Game-Theoretic Analysis | Evaluate the model's reasoning by engaging with it in a game-theoretic scenario. | Bertolazzi et al. (2023) |

Table 4: Interactive evaluation techniques designed to assess the model's reasoning behavior through dynamic interaction.

| Evaluation Method | Description | Exemplary Reference |
|---|---|---|
| Layer Probing | Probe the functional role of different layers of the model's architecture. | Pirozelli et al. (2023) |
| Attention Pattern Analysis | Analyze the model's attention matrices to gain insights into the underlying information flow within the model. | Hou et al. (2023) |
| Activation Patching | Alter specific neuron activations within a model and observe its impact on the model's output. | Dutta et al. (2024) |

Table 5: Overview of mechanistic evaluation methods that assess the model's reasoning by delving into its internal mechanisms throughout the reasoning process.

