# OpenReview forum: "Beyond Accuracy: Evaluating the Reasoning Behavior of Large Language Models - A Survey"
_colmweb.org/COLM/2024/Conference — COLM_

### Official Review · Reviewer_Ds6c · 2024-05-08

**Rating:** 6
**Confidence:** 3
**Ethics Flag:** 1

**Summary:**

This paper performs a survey on evaluating the reasoning behavior of LLM from two aspects:
1) the behavior of LLM on reasoning tasks.
2) the evaluation method of LLM reasoning.
Overall, the paper is well-written and includes the up-to-date literatures with respect to LLM reasoning. I think the practitioners will be interested in this paper and get insights from this paper about the research progress of LLM reasoning.

**Reasons To Accept:**

1) well organized related work to help the researchers quickly go through the progress in this domain.
2) LLM reasoning is important for LLM-based NLP, which is a hot research point.

**Reasons To Reject:**

1) The presentation need to be substantially improved. Long paragraphs of text discussion are tiring to read. It is difficult to quickly grasp the key findings or conclusions in publication survey. I think the authors should add more figures and tables as well as highlighted text to better understand.

---

> ### Author Rebuttal · Authors · 2024-05-29
>
> We express our gratitude to the reviewer for the thoughtful and constructive feedback on our survey which can “help the researchers [to] quickly go through the progress” in the domain of evaluating LLM-based reasoning behavior. We are especially grateful for the reviewer’s notion that “the practitioners will be interested in this paper and get insights from this paper about the research progress of LLM reasoning”.
>
> We would like to address the reviewer’s main concern regarding the presentation of our work:
>
> We appreciate the reviewer’s suggestion. We have made significant efforts to organize our survey well and present the key findings of the papers we review clearly. Additionally, we devoted considerable effort to the creation of additional supporting figures (see Figure 1 and Figure 2). We will add an additional table outlining the advantages and disadvantages of current evaluation methods (cf. rebuttal to reviewer bLrd), which is a suggestion by several reviewers and we agree will help enhance the presentation of our work.

---

> > ### Comment · Reviewer_Ds6c · 2024-06-01
> >
> > Thanks for the authors' responses. I have no questions and maintain my score.

---

### Official Review · Reviewer_tA8P · 2024-05-11

**Rating:** 6
**Confidence:** 3
**Ethics Flag:** 1

**Summary:**

The paper provides a comprehensive literature review on the evaluation of reasoning behaviors in large language models (LLMs). It effectively compiles findings from various studies, categorizing them by different reasoning tasks. Furthermore, the paper examines evaluation strategies for these behaviors, organizing them into four distinct methodological categories.

**Reasons To Accept:**

The survey is thorough and encompasses most aspects of reasoning behavior evaluation in the context of LLMs. This breadth ensures that it can serve as a valuable reference for researchers entering the field or looking to understand the current landscape of reasoning evaluation.

**Reasons To Reject:**

1. The framework for categorizing reasoning tasks is not particularly novel. Similar approaches have been detailed in existing literature, such as the HELM paper, which also evaluates models on both atomic and composite reasoning tasks. The choice to categorize reasoning behaviors according to specific tasks could be reconsidered. This is because certain erroneous behaviors manifest similarly across different tasks. For instance, an invalid entailment might represent a common type of reasoning error, yet it is characterized differently depending on the task at hand. A taxonomy based solely on tasks risks overlooking these cross-task similarities, which are crucial for deeper analytical insights.
2. The structure of the paper needs refinement to enhance its readability and impact. Key findings from the literature could be more effectively highlighted or presented in a more streamlined manner. As it currently stands, the paper's main points are somewhat obscured by overly lengthy and complex passages, making it difficult for readers to quickly grasp essential insights.

---

> ### Author Rebuttal · Authors · 2024-05-29
>
> We thank the reviewer for the thoughtful and constructive feedback.
>
> We are grateful for the reviewer's comments for a “comprehensive literature review” that is “thorough and encompasses most aspects of reasoning behavior evaluation in the context of LLMs”. Furthermore, we are pleased that the reviewer considers our survey “a valuable reference for researchers entering the field or looking to understand the current landscape of reasoning evaluation”.
>
> In the following, we would like to address the reviewer’s main concerns:
> 1. “The framework for categorizing reasoning tasks is not particularly novel” and “risks overlooking these cross-task similarities, which are crucial for deeper analytical insights.”
> - Our taxonomy is motivated by the literature we have reviewed, wherein most works employ tasks designed to capture specific types of reasoning. While similar categorizations might have been employed by existing works (e.g., the HELM paper), we believe that it remains helpful to systematically review literature that evaluates the reasoning behavior of LLMs. We acknowledge the reviewer’s point that similar behaviors might manifest across different tasks. However, we maintain that it is still possible to identify trends with the current taxonomy. For instance, across various task types, “slight alterations in the task context can markedly impair the models’ reasoning, multi-step reasoning is often reduced to shortcut pattern matching, and scenarios that deviate from the training distribution pose substantial challenges for current models” (see Section 3.3).
> 2. “Key findings from the literature could be more effectively highlighted or presented in a more streamlined manner.”
> - We tried our best to review the literature in a coherent manner, aiming not only to list the studies we surveyed but also to connect them through their findings. Nevertheless, acknowledging the diverse backgrounds of our readership, we appreciate your suggestion. We will try to present key findings more effectively in the final version of our paper.

---

> > ### Comment · Reviewer_tA8P · 2024-06-04
> >
> > Thanks for your reply. I have no further questions and maintain my score.

---

### Official Review · Reviewer_E7Vu · 2024-05-13

**Rating:** 6
**Confidence:** 4
**Ethics Flag:** 1

**Summary:**

This work summarizes the current reasoning works, including logical reasoning, mathematical reasoning, and causal reasoning. Firstly, they introduce the development of reasoning works. Then, they propose a taxonomy of evaluation methods to analyze the reasoning behaviors of LLMs. However, for the discussion part, there is no deeper insight into the models' reasoning processes. Since the paper is a survey evaluating the reasoning behavior of LLMs, it should summarize the current weaknesses of LLMs and propose some future directions.

**Questions To Authors:**

N/A

**Reasons To Accept:**

1. This paper reorganizes the current reasoning works and divides them into logical reasoning, mathematical reasoning, and causal reasoning.
2. This paper proposes a taxonomy of evaluation methods to analyze the reasoning behaviors of LLMs, which is missing in some existing surveys.

**Reasons To Reject:**

1. Lack of deep discussion. There is no deeper insight into the models' reasoning processes. They should summarize the current weaknesses of LLMs and propose some future directions.

---

> ### Author Rebuttal · Authors · 2024-05-29
>
> We express our gratitude for the reviewer’s feedback.
>
> We would like to address the reviewer’s main concerns:
>
> We agree that our discussion might be too succinct, and we intend to expand it in the final version of our paper (please refer to our rebuttal to reviewer AG6y). However, we would like to emphasize that our comprehensive review does indeed provide various insights into the models' reasoning processes. For example, slight alterations in the task context can significantly affect the models’ reasoning, multi-step reasoning is often reduced to shortcut pattern matching, and scenarios that differ from the training distribution pose substantial challenges for current models (see Section 3.3). In addition, we find that fundamental conceptual errors have been identified across various studies, which highlight the models’ deficiency in understanding basic principles in logic, mathematics and causal reasoning, especially in counterfactual setups.
>
> Moreover, we advocate for more sophisticated reasoning assessments as presented in Section 4, to better discern and comprehend the essential components missing in LLMs that are crucial for robust and systematic reasoning (see Section 5).
>
> We believe that these insights and recommendations are valuable contributions to the advancement of the research community.

---

> > ### Comment · Reviewer_E7Vu · 2024-06-06
> >
> > Thanks for the response. I will slightly improve the rating score. However, more discussion should be considered in the final version.

---

### Official Review · Reviewer_AG6y · 2024-05-22

**Rating:** 6
**Confidence:** 3
**Ethics Flag:** 1

**Summary:**

The paper is a review of recent literature of evaluating LLM reasoning. Authors focus specifically three families of core reasoning tasks: mathematical, logical and causal reasoning. After reviewing recent work, they discuss general trends in reasoning evaluation, arguing for paradigms that go beyond measuring the correctness of the outcome of a reasoning process and that also evaluate the reasoning process itself. They suggest reasoning capabilities of modern LLMs are not robust yet.

**Reasons To Accept:**

1. The paper addresses a topic at the very forefront of modern AI: LLM reasoning capabilities.
2. The paper surveys an impressive number of recent papers and summarizes them clearly.
3. The paper also presents a few insights on top of presenting a summary. Insights include: focus on outcome evaluation of most of prior work and general lack of robustness of LLM reasoning.

**Reasons To Reject:**

1. Some claims don't feel well-supported.
    1. I’m not sure I’m buying the distinction between reasoning behavior vs final task performance. You can still study reasoning behavior by tracking final task performance if you design your tasks really well such that they track particular kind of error or behavior. For instance, one can measure the compositionality gap [1] (the accuracy difference between two one-hop and a single two-hop question) to study a sophisticated reasoning behavior (two-hop reasoning) while relying on metrics tracking just an outcome of a well-designed task.

   2. How do authors motivate defining core reasoning tasks a logical, mathematical or causal reasoning? Why are they non-overlapping (e.g. doesn’t math reasoning pre-supposed logical reasoning)? Why do they exhaust the space of reasoning types? Should this typology be thought as listing natural kinds, carving nature at its joints or is it just a useful albeit fuzzy set of labels that LMs might not might not respect?

   3. Authors frequently hint at the brittleness of reasoning capabilities being a result of memorization, but don’t discuss this in detail and more rigorously. The strong reading of this claim is clearly false: LLMs do generalize to unseen problems, e.g. the gsm1k paper [2] finds that only some models (Mistral) show overfitting to gsm8k that could explain their good performance. I wonder what’s the more nuanced reading of the claim authors hint at. I’d be great to see it discussed in more detail.

2. The survey seems to be missing some important thread of recent work and is somewhat shallow when discussing some others.
    1.  This is less important, but some threads feel missing to me:
         1. Performance boosts from techniques for eliciting reasoning capabilities, e.g. chain of thought, self-ask, using feedback; the gap between externalized reasoning and zero-shot reasoning performance. Some relevant papers:
            1. Do Large Language Models Latently Perform Multi-Hop Reasoning?
            2. Large Language Models Cannot Self-Correct Reasoning Yet
        2. Faithfulness of externalized reasoning:
            1. Language Models Don't Always Say What They Think: Unfaithful Explanations in Chain-of-Thought Prompting
            2. Measuring Faithfulness in Chain-of-Thought Reasoning
        3. The role of knowledge representation in multi-hop reasoning
            1. Physics of Language Models: Part 3.2, Knowledge Manipulation
            2. Taken out of context: On measuring situational awareness in LLMs
   2. I’m a bit worried that the authors paint their picture relying on some pre-2021 papers conducting experiments on GPT-3, RoBERTa and T5 or even GPT-2 and BART! I think modern frontier LLMs (such as Claude 3 Opus, GPT-4o) are significant more capable than GPT-3-series models and a lot of findings of prior work don’t translate to them.
   3. Section 4.4 feels very blunt: “deeper insights into the model’s operational logic can be gained” is a cliche, please tell us what we’ve learned exactly.

Some more specific comments:

> skeptics contend that the models’ performance is a mere reflection of the extensive training data and their vast number of parameters (Razeghi et al., 2022; Wu et al., 2023; Berglund et al., 2024).

I don’t think Berglund et al., 2024 lends evidence to this claim. The reversal curse is a failure of knowledge storage/retrieval not reasoning.

Moreover, I’d rephrase the claim to make it more precise. That reasoning capability is a result of increased scale is not a skeptical take at all. We all know that larger models and models trained on more data generally tend to get better results. Probably what the authors wanted to say is something about memorization vs generalization?

> Furthermore, theoretical findings suggest that due to the models’ autoregressive nature, mistakes in early stages of the reasoning process can lead to compounding errors that exponentially diminish the likelihood of arriving at the correct solution in subsequent steps.

Citation needed. Maybe [3], section 4?

> At the same time, McKenzie et al. (2023) demonstrate a form of goal misgeneralization

I’d avoid the term “goal misgeneralization” as it has its own technical meaning [4]

> extensive training data, rather than genuine reasoning capabilities

This strikes me as a false dichotomy. We don’t have any reason to think that genuine reasoning capabilities cannot be learned from extensive training data. What the authors mean is perhaps memorization vs generalization again.

> Similarly, McKenzie et al. (2023) indicate that LLMs tend to replicate
human-like logical errors when engaging with the logical principle of modus tollens, suggesting an imitation of flawed reasoning patterns from their training data. Notably, this trend becomes more pronounced with increasing model size, a phenomenon denoted as *inverse scaling*.

This is too strong Some tasks show inverse scaling, but it’s a minority and tasks in McKenzie et al. (2023) were selected for exhibiting inverse scaling. For the vast majority of reasoning tasks, performance improves with scale.

[1] https://arxiv.org/abs/2210.03350

[2] https://arxiv.org/abs/2405.00332

[3] https://arxiv.org/pdf/2305.18654

[4] https://arxiv.org/abs/2210.01790

---

> ### Author Rebuttal · Authors · 2024-05-29
>
> We thank the reviewer for the insightful feedback.
>
> We appreciate the recognition of the “impressive number of recent papers” covered in our survey and are encouraged that these papers are summarized “clearly.” Additionally, we are pleased the reviewer highlights some of our paper’s “insights on top of presenting a summary.”
>
> Due to character limitations, we will address the reviewer's main concerns and respond to specific comments in a separate rebuttal.
> 1. Reasoning behavior vs. task performance:
> - We agree that for more sophisticated task designs, conclusions about the model’s reasoning behavior can be drawn based on the model’s task performance, particularly when tasks are designed to track specific types of errors. In Section 4.1, we note that this: “outcome-oriented approach […] can nonetheless provide insights into the model’s reasoning behavior. For instance, a thorough error analysis can unveil conceptual errors”. We would like to clarify that our critique is directed at benchmarks with less nuanced task designs where accuracy metrics provide a less informative or even misleading assessment of the models’ reasoning \[1]. Thanks for raising this point as it helps us to clarify our stance.
> 2. Categorization of reasoning tasks:
> - First, we would like to emphasize that the tasks and abilities listed in our paper are not exhaustive, as noted in the caption of Figure 1. Our taxonomy is motivated by the literature we have reviewed, wherein most works employ tasks designed to capture a particular type of reasoning. Although these tasks may invoke more than one reasoning skill, this taxonomy helps us to systematically review the reasoning behavior of LLMs within the context of such tasks.
> 3. Robustness of LLM-based reasoning:
> - We agree with the reviewer that our discussion is concise, and we plan to extend it in the final version of our paper. One significant challenge in evaluating the reasoning capacities of LLMs is the closed-source nature of their training data. Various studies indicate that current LLMs are significantly affected by benchmark leakage \[1,2]. This underscores the need for more nuanced evaluations that move beyond potentially misleading accuracy values. While LLMs can generalize to unseen samples, our review highlights several studies that reveal conceptual errors and limitations in novel reasoning scenarios, which account for data leakage
>
> \[1]: https://aclanthology.org/2024.eacl-long.5
>
> \[2]: https://doi.org/10.48550/arXiv.2311.01964

---

> > ### Comment · Reviewer_AG6y · 2024-06-03
> >
> > Thanks for the response! Where can I find that separate rebuttal in which you respond to my specific comments?

---

> > > ### Author Response · Authors · 2024-06-04
> > > **Response to Reviewer AG6y**
> > >
> > > Thank you for the response. Please find our answer to the more specific comments below.
> > >
> > > Missing thread of work.
> > > 1. We appreciate the reviewer's suggestions regarding interesting aspects of LLM-based reasoning. However, we have already reviewed a substantial number of recent papers and believe that incorporating the reviewer's suggestions would expand the scope of our survey beyond its intended focus.
> > > 2. We have reviewed very recent studies up to March 2024 (the month of submission) and have reported various findings for advanced models such as GPT-4, PaLM 2, and Claude 2. Additionally, we consistently specify which LLM was utilized in each respective study.
> > >
> > > Regarding the reviewer’s more specific comments:
> > > 1. The reversal curse indicates that models struggle to capture a basic symmetry property of the identity relation, instead learning a more superficial, unidirectional relationship. As noted by the authors, this is “a basic failure of logical deduction in the LLM’s training process”, which could extend to other relationships that are essential for effective reasoning such as “logical implications [...], spatial relationships [...], or n-place relations.” (see Section 1 and 4.1 of [Berglund et al. 2024](https://doi.org/10.48550/arXiv.2309.12288))
> > > 2. The mentioned paragraph relates to the previous sentence discussing the work of [Dziri et al. (2023)](https://doi.org/10.48550/arXiv.2305.18654).
> > > 3. The term "goal misgeneralization" is used by [McKenzie et al. (2023)](https://doi.org/10.48550/arXiv.2306.09479) and by us in the same context as defined in reference [[4]](https://doi.org/10.48550/arXiv.2210.01790). Could the reviewer please provide further clarification?
> > > 4. The limited number of tasks in [McKenzie et al. (2023)](https://doi.org/10.48550/arXiv.2306.09479) does not invalidate the authors' finding that LLMs replicate certain types of human-like logical errors present in their training data.

---

> ### Comment · Reviewer_AG6y · 2024-06-04
>
> Thanks for the response!
>
> > The reversal curse indicates that models struggle to capture a basic symmetry property of the identity relation, instead learning a more superficial, unidirectional relationship. As noted by the authors, this is “a basic failure of logical deduction in the LLM’s training process”, which could extend to other relationships that are essential for effective reasoning such as “logical implications [...], spatial relationships [...], or n-place relations.” (see Section 1 and 4.1 of Berglund et al. 2024)
>
> I disagree. The reversal curse does not happen in context: models are perfectly capably of inferring "Joe Biden is US president" when **prompted** with "US president is Joe Biden"; reversing relations like this is not a problem. They just struggle to do that during in-weights learning. This suggests it's a failure of knowledge storage/retrieval (when a "A is B" is stored in one direction, models fail to retrieve A when conditioned on B) not of logical deduction.
>
> The first passage you cited is a hypothetical that is debunked two sentences later: "Moreover, this is not explained by the LLM not understanding logical deduction." The second cited passage is also a question, not a claim.
>
> > The limited number of tasks in McKenzie et al. (2023) does not invalidate the authors' finding that LLMs replicate certain types of human-like logical errors present in their training data.
>
> They *sometimes* replicate human errors. I think asserting this without a "sometimes" is a claim the McKenzie et al. don't provide sufficient evidence for.
>
> > The term "goal misgeneralization" is used by McKenzie et al. (2023) and by us in the same context as defined in reference [4]. Could the reviewer please provide further clarification?
>
> You're right!

---

> > ### Author Response · Authors · 2024-06-06
> > **Response to Reviewer AG6y**
> >
> > We appreciate the reviewer’s engaging discussion on the reversal curse.
> >
> > This phenomenon can be addressed from two perspectives: knowledge retrieval/storage or logical deduction. While we agree that the in-context learning experiments of [Berglund et al., 2024](https://doi.org/10.48550/arXiv.2309.12288) suggest that the reversal curse might stem from a failure of information retrieval, we respectfully disagree that this phenomena is unrelated to the model’s reasoning. As noted by the authors, while attributing the reversal curse solely to logical deduction may be considered "a simplification," it remains a "useful" perspective to relate it to deductive reasoning. Specifically, the reversal curse illustrates “a basic inability [of LLMs] to generalize beyond the training data” (see Section 1), which will have a significant impact on their reasoning behavior.

---

> > > ### Comment · Reviewer_AG6y · 2024-06-06
> > >
> > > Thanks for the response. Let's agree to disagree then! I think it's perfectly fine to present an interpretation of the reversal curse as a reasoning failure in the paper, but it should be clear it's a novel interpretation put forth by the authors of the present paper and it should be supported by some arguments.
> > >
> > > > Specifically, the reversal curse illustrates “a basic inability [of LLMs] to generalize beyond the training data” (see Section 1), which will have a significant impact on their reasoning behavior.
> > >
> > > I agree it will impact reasoning behavior but IMHO it's not justified to call it a reasoning failure. If a model training data cutoff is 2023, the model won't be able to reason about the gold and silver medalists of 2024 Summer Olympics. But we wouldn't call that a reasoning failure, would we?

---

### Official Review · Reviewer_qouy · 2024-05-22

**Rating:** 7
**Confidence:** 3
**Ethics Flag:** 1

**Summary:**

This paper provides a comprehensive review of literature that evaluates LLM-based reasoning beyond mere task accuracy, which covers three core reasoning tasks, i,e,, logical, mathematic, and causal reasoning, as well as current evaluation frameworks. The review is clear, and helpful to the researchers in this community.

**Questions To Authors:**

In the conclusion, the authors highlighted the necessity for more nuanced analyses of the models’ reasoning behavior, especially in novel scenarios, and how about the evalution of the capability of the LLMs in few-shot learning scenarios? it will be nice to provide analysis of such work.

**Reasons To Accept:**

1）The review is comprehensive, covering the reasoing tasks, evaluation frameworks, as well as some insights from the evaluation of LLMs.

2) The review provides a clear picture of the research in evaluation of the LLM, especially the two figures in the paper are of great help to understand the situation.

**Reasons To Reject:**

1) The review lacks some concrete examples to specify the advantage and disadvantage of the evaluaiton frameworks.

2) Not sure about whether such reviews fall in the scope of the conference.

---

> ### Author Rebuttal · Authors · 2024-05-29
>
> We thank the reviewer for the constructive and favorable feedback.
>
> We highly appreciate that the reviewer acknowledges our efforts to present a “comprehensive review of literature that evaluates LLM-based reasoning beyond mere task accuracy”, which “provides a clear picture” and is “helpful to the researchers in this community”. Furthermore, we are encouraged that the reviewer finds that “the two figures in the paper are of great help to understand the situation”.
>
> We would like to address the reviewer’s main concerns:
>
> 1. Advantages and disadvantages of each evaluation framework:
>
> - We agree that an overview of the advantages and disadvantages for each evaluation framework would benefit the research community, and acknowledge that this is currently missing in our survey. Therefore, we have created a respective overview table (please refer to our rebuttal to reviewer bLrd), and are happy to include it in the final version of our paper. If you have further suggestions on the summary above please let us know.
>
> 2. Scope of the conference:
>
> - The conference's FAQ page indicates: "We don't have a special track for survey papers. If you wish, you can submit a survey paper and we will route it to the appropriate area chairs and reviewers and based on their judgment, you will get a decision". Based on this information, along with various announcements on Twitter/X (cf. [this post](https://twitter.com/infoxiao/status/1757516389617996151)), we are confident that COLM welcomes survey papers.
>
> In response to the reviewer's question, could the reviewer kindly provide further clarification? Is the question targeted at how LLMs behave in reasoning scenarios when prompted via in-context learning?

---

### Official Review · Reviewer_bLrd · 2024-05-24

**Rating:** 6
**Confidence:** 4
**Ethics Flag:** 1

**Summary:**

This paper is a survey of reasoning behavior in large language models. It seeks to answer 2 research questions: 1) how do current large language models behave in mathematical, causal and logical reasoning tasks, and 2) what are the evaluation methods for assessing the reasoning capabilities of large language models. The survey finds that LLMs rely on pattern matching to reasoning examples in their training distribution, leading to them making conceptual errors in reasoning particularly in out-of-distribution scenarios. To better assess true reasoning behavior, the survey categorizes evaluation methods into conclusion-based, rationale-based, interactive and mechanistic evaluations.

The paper is a well-researched summary of assessments of reasoning behavior in large language models. By using a taxonomy of logical (deductive, inductive, abductive), mathematical, and causal reasoning tasks, the paper is able to capture a good breadth of the literature, and serves as a good overview of the reasoning abilities and deficiencies of current large language models. The writing is clear and easy to follow, and is often able to capture the main ideas of cited papers in a succinct manner. This is a noteworthy contribution to a community that is contending with an open question of how much reasoning abilities truly emerge at scale, or if LLMs are just "stochastic parrots" of their data.

That said, the paper could have gone further in its contribution to the debate. As it stands, the survey is opinionated in putting forth a coherent argument for LLMs being a reflection of their training distribution by highlighting deficiencies in reasoning behavior. There could have been further analysis on why eliciting reasoning behavior (chain-of-thought, tree-of-thought style or the ReAct framework) tend to improve reasoning outcomes, as at least at first glance this isn't immediately suggestive of pattern matching. In fact, a taxonomy on such methods to induce reasoning ability would lead to a more comprehensive survey; while the paper is more focused on evaluation methods, there seems to have been a missing component of how that reasoning behavior is obtained (via prompting, etc.) which is a first-order discussion.

Moreover, Section 4 (taxonomy of evaluation methods) could have gone further in discussing 1) the overall recent trends for the area of reasoning evaluation, 2) why the field has evolved as such, and 3) the tradeoffs using each evaluation method in the taxonomy. This would go a long way in providing a suggested direction for future work in improving our methodology for assessing reasoning capabilities.

**Questions To Authors:**

1. Do the authors agree that a section on how reasoning behavior is induced from LLMs (i.e. why reasoning outcomes are better upon prompting for rationales, or structuring thoughts in a particular form in prompts) is important? If not, why not?
2. Upon review of the literature surrounding evaluation methods for reasoning behaviors, for each evaluation method can the authors provide a) a synthesis of the tradeoffs for that method, and b) suggestions for future work addressing the limitations of that evaluation method
3. (comment) Section 3.1.1 includes an explanation of mechanistic evaluation as part of the review on behavior in deductive reasoning tasks, but this is later largely duplicated in Section 4.4. This seems redundant, or could be better structured to avoid duplication.

**Reasons To Accept:**

1. The paper is a clearly written and well-researched summary of the deficiencies of reasoning behavior in LLMs, as well as of the different evaluation methodologies for assessing reasoning behavior. In particular, the paper does a good job of succinctly presenting the main ideas of papers it cites as part of the survey.
2. Reasoning in LLMs is an important field, and a survey of the literature is helpful for the community. The topic is also very appropriate for COLM.

**Reasons To Reject:**

1. The paper does not synthesize trends or provide suggestions for future work in evaluating reasoning abilities of LLMs.
2. The paper is missing some discussion on how reasoning behavior is elicited from LLMs, which seems important for such a survey paper.

---

> ### Author Rebuttal · Authors · 2024-05-29
>
> We thank the reviewer for the thoughtful feedback.
>
> Firstly, we wish to express our gratitude for acknowledging our efforts in providing a “well-researched summary”, which captures “a good breadth” of literature, “and serves as a good overview of the reasoning abilities and deficiencies" of current LLMs. We are encouraged that the reviewer finds our writing “clear and easy to follow”, and are grateful for the recognition of our work as a “noteworthy contribution to a community that is contending with an open question of how much reasoning abilities truly emerge at scale”.
>
> In response to the reviewer’s questions
> 1. How reasoning behavior is induced:
> - Our survey focuses on studies that evaluate the reasoning behavior of LLMs beyond mere task accuracy. We agree with the reviewer on the importance of linking observed reasoning behavior with the employed reasoning techniques. Consequently, we always indicate when a technique other than standard prompting is used. However, we believe that providing an overview of reasoning techniques and their effects on model reasoning would expand the scope of our survey. Additionally, existing surveys already offer such overviews. For reference, please see [[1](https://doi.org/10.18653/v1/2023.findings-acl.67),[2](https://aclanthology.org/2023.acl-long.294)] as mentioned in Section 1 of our paper.
>
> 2. Trade-offs for each evaluation method:
> - We agree and thank you for this suggestion. Below, we summarize the main trade-offs for each evaluation method. We intend to include an elaborate comparison in the final version of our paper.
>
> ||Advantages|Disadvantages|
> |-|-|-|
> |**Conclusion-based evaluation**|Allows for controlled setups|Limited insights|
> ||Provides metrics for comparison|Less reliable (cf. unfaithful reasoning)|
> ||Easy to automate and scale||
> ||Easy to reproduce||
> ||||
> |**Rationale-based evaluation**|Offers more nuanced insights|Difficult to automate and scale|
> ||More robust in certain scenarios [[3](https://doi.org/10.48550/arXiv.2404.08382)]|Might require expert interpretation|
> ||||
> |**Interactive evaluation**|Highly flexible|Expensive|
> ||Customizable to model behavior|Difficult to automate and scale|
> |||Less standardized and reproducible|
> ||||
> |**Mechanistic evaluation**|Identifies input features or model circuits responsible for specific behavior|Compute-intensive|
> ||Supports direct interventions on model internals|Findings may not generalize across tasks or models|
> |||Results may be hard to interpret|
> ||||

---

> > ### Comment · Reviewer_bLrd · 2024-06-06
> >
> > Thank you for the authors' response. I acknowledge that giving an overview of reasoning techniques expands the scope of the survey, but I do encourage the authors to give their taxonomy of the techniques in the appendix of the final paper if accepted.
> >
> > I maintain my view that a synthesis of the tradeoffs for evaluation methods and suggestions for future work are critical for the narrative of such a survey paper. I appreciate the authors' summary of the tradeoffs, which seem sensible to me. I will maintain my score for now, but the paper would deserve a 7 if each section on evaluation techniques ends with this summary and provides the community with research ideas to pursue to address the disadvantages of the respective techniques.

---

> > > ### Author Response · Authors · 2024-06-07
> > > **Response to Reviewer bLrd**
> > >
> > > We appreciate the reviewer's valuable feedback and suggestions.
> > >
> > > We agree that such a synthesis would be beneficial to this survey and are committed to incorporating it into the final version of our paper.

---

### Decision · Program_Chairs · 2024-07-10

**Decision:**

Accept

**Comment:**

This paper provides a review of the reasoning abilities of LLMs and the prevalent approaches for evaluating these reasoning abilities.

The reviewers highlighted several strengths of this paper: it provides clear and succinct summaries of a wide range of papers, its topic is important and timely, the taxonomy of evaluation types that it provides is useful, and it provides some clear high-level conclusions and takeaways rather than simply listing papers (e.g., summarizing the strengths and weaknesses of evaluation types, and giving a big-picture view of LLM reasoning abilities).

Limitations that reviewers noted included the fact that some relevant strands of literature were not discussed (particularly prompting approaches for improving LLM reasoning, such as CoT), imprecision in some claims (particularly what is meant by brittleness and memorization), and some presentational shortcomings (i.e., insufficient highlighting of key points and takeaways).

Despite these limitations, I believe that the paper will be a useful one that will be of interest to a large subset of the COLM audience. Therefore, I am recommending acceptance. I encourage the authors to make the following revisions to the paper (in the camera-ready version if it is accepted to COLM, or in the version that the authors submit elsewhere if it is not accepted):
- Adding the table that the authors discuss in the response to Reviewer bLrd; this would strengthen the paper and help address some of these limitations.
- Adding some discussion of prompting approaches that improve reasoning performance such as CoT, ReAct, etc. The fact that this suggestion came up in two reviews suggests that many readers will have this reaction too. Therefore, even if you view it as out-of-scope, it seems worthwhile to add some discussion of it - even if that discussion simply states that these directions are out-of-scope.

[At least one review was discounted during the decision process due to quality]